# Efficacy of Pancreatic Endotherapy in Pancreatic Ascites and Pleural Effusion

**DOI:** 10.3390/medsci5020006

**Published:** 2017-03-27

**Authors:** Sudhir Gupta, Nitin Gaikwad, Amol Samarth, Niraj Sawalakhe, Tushar Sankalecha

**Affiliations:** Department of Gastroenterology, Government Medical College and Super Specialty Hospital, Nagpur, Maharashtra 440003, India; sudhirjgupta@gmail.com (S.G.); amolsamarth2012@gmail.com (A.S.); sawalakhe_1@yahoo.com (N.S.); dr.tushar30@gmail.com (T.S.)

**Keywords:** ductal leak, pancreatogram, ascites

## Abstract

Pancreatic ascites and effusion is a challenging complication to manage, hence our aim was to evaluate the efficacy of pancreatic endotherapy in pancreatic ascites and pleural effusion. Endotherapy included endoscopic retrograde cholangiopancreatography (ERCP) with a pancreatogram and pancreatic stent placement across the leak in patients with pancreatic ascites/effusion. A total of 53 patients were included after successful cannulation. The male:female ratio was 7.8:1. The pancreatogram revealed a leak from the pancreatic duct in 20/53 (37.73%) patients. The most common leak site was the pancreatic body in 10/53 (18.9%) patients followed by the tail in 6/53 (11.32%) patients and the genu in 4/53 (7.5%) patients. In 29/53 (54.7%) patients, stent was placed beyond the leak site. Sphincterotomy was done in 7/53 (13.2%) patients, and in five patients with an obscure leak site, stent was placed empirically. A total of 39/53 (73.6%) patients benefited in terms of achieving the complete resolution of ascites and pleural effusion. The factors which were significant for the success of pancreatic endotherapy in the multivariate analysis were the site of the pancreatic ductal leak (*p* value = 0.008) and the ability of the stent to cross the leak site (*p* value = 0.004). To sum up, bridging the pancreatic ductal leak by stent offers a high rate of success. Pancreatic endotherapy is less invasive and highly effective in managing pancreatic ascites/pleural effusion.

## 1. Introduction

Pancreatic diseases and their complications are common in the Indian Subcontinent. Common causes of pancreatic diseases include gall stone disease, alcohol and tropical pancreatitis. Long-term complications of severe pancreatitis progress to chronicity with intraductal stones, pancreatic ductal strictures, distal biliary strictures and pseudocysts. Pancreatic ascites and pleural effusion are among the least common complications of pancreatitis. The exact prevalence of the pancreatic ascites is not known. Estimated prevalence is 3.5% in patients with chronic pancreatitis and 6%–14% in patients with pseudocyst [1]. The mechanisms of the pancreatic ascites include pancreatic ductal leak, internal pancreatic fistula, pseudocyst rupture and trauma to the pancreas [2]. Although it is the least common complication, it carries more grievous prognosis. Management options include octreotide therapy, nutritional therapy, endoscopic therapy and surgery. Pancreatic endotherapy offers many advantages over other forms of therapy in terms of comfortability and excellent response rate. It has achieved a high success rate and low morbidity in properly selected patients. Pancreatic endotherapy is more affordable when compared to surgery; this factor is of great importance in poor-resource countries such as India where this problem is highly prevalent. Hence, we carried out this study to analyze the efficacy of pancreatic endotherapy in patients with pancreatic ascites and pleural effusion.

## 2. Materials and Methods 

Institutional ethics committee approval was obtained before the study. The approval code is 695EC/Pharmac/GMC/NGP dated 5 June 2013. All patients with pancreatic ascites and pleural effusion were seen in the Department of Gastroenterology, Government Medical College and Super Specialty Hospital (SSH), Nagpur, Maharashtra, India. The duration of the study was 3 years from July 2013 to June 2016. The retrospective data was reviewed prospectively for the preparation of this manuscript.

### 2.1. Inclusion and Exclusion Criteria

Inclusion criteria: aged ≥18 years; pancreatic ascites and pleural effusion for 4 weeks; failed conservative line of management in the form of octreotide; antibiotics and nasojejunal feeding for at least 4 weeks; persistent ascites and pleural effusion after one or more paracenteses; fluid amylase level of >1000 IU/dL; evidence of acute or chronic pancreatitis on imaging study (computed tomography (CT), ultrasonography (USG), magnetic resonance imaging (MRI)); absence of other causes of ascites.

Exclusion criteria: aged <18 years; minimal ascites or pleural effusion; receiving treatment for other indications such as endoscopic management of pancreatic pain, or transmural drainage of pseudocysts.

### 2.2. Analyses

All patients underwent the following analyses: Complete blood count; liver function test; renal function test; blood sugar test; prothrombin time/international normalised ratio (PT/INR); ultrasound of abdomen; ascitic fluid analysis including amylase levels, and upper gastrointestinal endoscopy; CT scan of the abdomen to determine the status of the pancreas, pseudocysts and pancreatic duct dilatation. Magnetic resonance cholangiopancreatography (MRCP) was only done in selected patients due to financial constraints. Secretin-enhanced MRCP was not done in our study due to non-availability. In the endoscopic retrograde cholangiopancreatography (ERCP) setting, pancreatic ductal cannulation was achieved with 0.032” guidewire under direct fluoroscopic vision. A pancreatogram was taken to determine the presence and site of the leak (Figure 1) along with associated stones and strictures.

### 2.3. Procedure

After determining the leak site, plastic pancreatic stent was placed across the leak. Plastic pancreatic stents—either straight end or single pigtail stent of size 5 Fr × 5 cm, 5 Fr × 7 cm, 7 Fr × 5 cm—were used depending on the leak site. Nasojejunal tube feeding was started. Pancreatic stent was removed after 4–6 weeks depending on the resolution time of ascites/pleural effusion. Patients who underwent endotherapy did not receive parenteral nutrition, or octreotide therapy after stent placement. Communicating pseudocysts were drained by transpapillary stenting. Those patients in whom the leak site was not identified i.e., obscure, were empirically stented with 5 Fr stent. Successful pancreatic endotherapy was defined as resolution of ascites and pleural effusion at 6 weeks—evident clinically and by X-ray/ultrasonography. Failure of endoscopic therapy was defined as persistence of fluid or partial resolution of ascites and pleural effusion at 6 weeks (Figure 2).

### 2.4. Statistical Analysis

The statistical analysis performed in the present study is descriptive and inferential. Continuous measurements are presented as mean ± standard deviation (SD) and categorical measurements are presented as a number (%). The significance of study parameters on a categorical scale between two or more groups was done by the Chi-square/Fisher Exact test. The significance of study parameters on a continuous scale between two groups (inter-group analysis) on metric parameters was made by the Student’s *t*-test. Multivariate logistic regression analysis was employed to find out the independent factors affecting the outcome of pancreatic endotherapy at 6 weeks. 

Data analysis was performed using the statistical software SAS 9.2, SPSS 15.0, Stata 10.1, MedCalc 9.0.1, Systat 12.0 and R environment ver.2.11.1. (SAS Institute Inc. 100 SAS Campus Drive, Cary, NC, USA).

## 3. Results

A total of 53/56 patients enrolled after successful cannulation. The male:female ratio was 7.8:1 (Table 1). The mean age was 43.89 ± 8.1 years. A total of 11/53 (20.8%) patients had acute pancreatitis and 43/53 (79.2%) had chronic pancreatitis. The most common cause of the pancreatitis in our study was alcohol in 42/53 (79.2%) patients. Other etiologies which we found were idiopathic pancreatitis in 7/53 (13.2%) patients and traumatic pancreatitis in 4/53 (7.5%) patients. Ascites alone was present in 37/53 (69.8%) patients, pleural effusion alone was present in 5/53 (9.4%) patients, and ascites with pleural effusion was present in 11/53 (20.8%) patients. One or more pseudocysts were documented in 28/53 (52.83%) patients. Figure 2 shows leak and communicating pseudocyst as a group in 29 patients. Twenty of these patients had a direct leak on the pancreatogram while nine patients had a communicating pseudocyst. A non-communicating pseudocyst was present in 19 patients. The pancreatogram revealed a leak from the pancreatic duct in 20/53 (37.73%) patients. The most common leak site was the body in 10/53 (18.9%) patients followed by tail in 6/53 (11.32%) patients and genu in 4/53 (7.5%) patients (Table 2). The leak site was not documented i.e., obscure in 5/53 (9.43%) patients. In 29/53 (54.7%) patients, stent was placed beyond the leak which also includes patients with communicating pseudocysts. In 12/53 (22.6%) patients, stent was not passed beyond the leak. Sphincterotomy was done in 7/53 (13.2%) patients, and in five patients with an obscure leak site, stent was placed empirically. These patients were followed for a period of 6 weeks and reassessed for ascites and pleural effusion. A total of 39/53 (73.6%) patients benefitted from the pancreatic stenting in terms of achieving the complete resolution of ascites and pleural effusion. A total of 14/53 (26.4%) patients failed to respond to the endoscopic therapy at 6 weeks. A total of 6/53 (11.3%) patients died within 6 weeks of enrollment due to the progression of underlying ascites and complications such assepsis and bleeding, which occurred in two patients. Clinical presentation, type of pancreatitis, etiology of pancreatitis, leak site and stent crossing across the leak were compared with the outcome of pancreatic endotherapy. The factors which were significant for the success of pancreatic endotherapy on multivariate analysis were leak site (*p* value = 0.008) and stent crossing the leak site (*p* value = 0.004) (Table 3).

## 4. Discussion

Smith was the first to describe pancreatic ascites in patients with chronic pancreatitis [3]. A pancreatic ductal leak usually occurs as a complication of severe acute pancreatitis or underlying long standing chronic pancreatitis. The pathogenesis of ductal disruption leading to ascites formation includes pancreatic necrosis, severe inflammation or obstruction of the duct, rupture of a pseudocyst into the peritoneal cavity and relentless progression of chronic pancreatitis [2]. The most commoncause of a leak in chronic pancreatitis is a pseudocyst in 80% of cases and direct disruption of the ductin the remaining 20% of cases [4]. Commonly used imaging modalities for pancreatic diseases are CT scan and MRCP which provide better delineation of the pancreatic duct. The advent of helical computed tomography has allowed us to detect contiguous fluid collections and any upstream dilation behind a stricture or stone. Further, imaging with MRCP allows us to take a pancreatogram and administer secretin to define the exact site of the disruption [5].

Management of patients with pancreatic ascites/effusion is challenging because of the various associated comorbidities. In addition to this, patients are usually malnourished due to recurrent vomiting, poor oral intake, diabetes mellitus and disease related factors like multiple strictures and large pancreatic ductal calculi. Treatment modalities for the pancreatic ascites include conservative medical approach, surgical treatment and endoscopic pancreatic therapy. Conservative treatment includes total parenteral nutrition, drugs to decrease the pancreatic secretions like octreotide, somatostatin, and IV fluids [6]. Conservative therapy for a month has a success rate of 25%–60% [7]. According to Cameron et al ascites subsides in 17%–50 % of patients with conservative management [8]. Disadvantages of Conservative approach are that it is expensive, requires prolonged hospitalization, and mortality rates range from 1%–25% [9]. Surgical management is indicated in patients with intracystic bleed, complete obstruction of pancreatic duct and failure of other modalities of treatment. Surgical management includes either pancreatic resection combined with ductal drainage procedure or enteropancreatic anastomosis [10]. Surgical treatment has many complications and the occurrence of death in 1%–20% [11]. Localization of site of leak is essential for planning the therapy.

Pancreatic endotherapy is the major form of endotherapy, and includes pancreatic sphincterotomy and transpapillary pancreatic stenting across the leak site. This will decrease the pressure gradient at the pancreatic sphincter and will allow the flow of pancreatic secretions along a low resistance path to the duodenum, which subsequently heals the site of the ductal leak [11]. In our study, around 39/53 (73.6%) patients benefited from the pancreatic endotherapy treatment. Eckhauser et al. reported the efficacy of endotherapy in 50%–90% of patients [12]. Factors which favour the healing of the ductal leak are absence of stones, strictures and the ability to place the stent across the leak site. In our study, the most common leak site was the body of pancreas which, when stented, has a high success rate of 90% when compared with the tail leak for which the success rate was 10%; this could be explained by the inability of stent to cross the site of the leak in the tail region. The length of the pancreatic stent depends on the leak site; usually, leaks in the head and body require 5 cm and 7 cm stents and more proximal leaks in the tail will require longer length stents. Although stent therapy has been reported to be more invasive and carries a greater risk than medical therapy, it is safer than surgical treatment [13]. In our study, six patients died within 6 weeks of enrollment; this was due to the progression of an underlying disease in four patients and two patients had sepsis and bleeding. There was no post-ERCP pancreatitis in our series, which could be due to the smaller number of patients in our series. In the study of Bhasin et al. [14], endoscopic therapy is successful in closing the leaks in 60% of patients. Factors which were associated with a better outcome include a partial disruption, successfully bridging the disruption with a stent and longer duration of stent placement. Pancreatic ascites and pleural effusion can be cured by endotherapy and transpapillary nasopancreatic drainage [14]. Endotherapy offers many distinct additional advantages over surgery. The procedure can be repeated with less risk, unlike the morbidity and difficulty associated with repeat surgery. Absence of an internal control group can be seen as a limitation of our study. However, our prime aim was to share our experience regarding the efficacy of pancreatic endotherapy, how many patients benefited and what were the factors for the difference in the outcome of endotherapy such as the leak site and the ability to bridge the leak by stenting.

## 5. Conclusions

Pancreatic endotherapy is a safe and effective modality of treatment in pancreatic ascites and pleural effusion with a significant reduction in morbidity and mortality. Pancreatic ductal stenting can be used as a first line management in such patients. The ability of endotherapy to find the leak site and bridge this leak site with pancreatic stenting are the important independent factors which predict the resolution of pancreatic ascites and effusion.

## Figures and Tables

**Figure 1 medsci-05-00006-f001:**
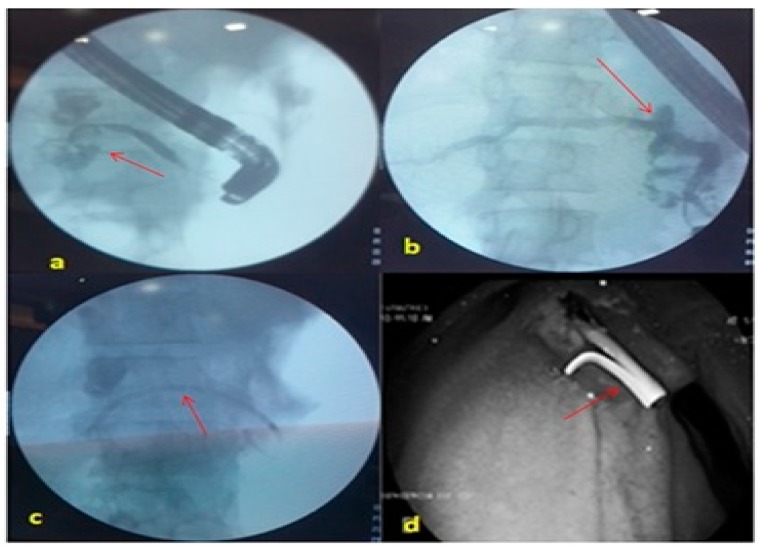
Pancreatogram fluoroscopic images (**a**) Pancreatogram showing a leak in the midbody of the pancreas (arrow); (**b**) Pancreatogram showing a leak in the head region of the pancreas; (**c**) Plastic stent across the leak; (**d**) Distal end of pancreatic stent.

**Figure 2 medsci-05-00006-f002:**
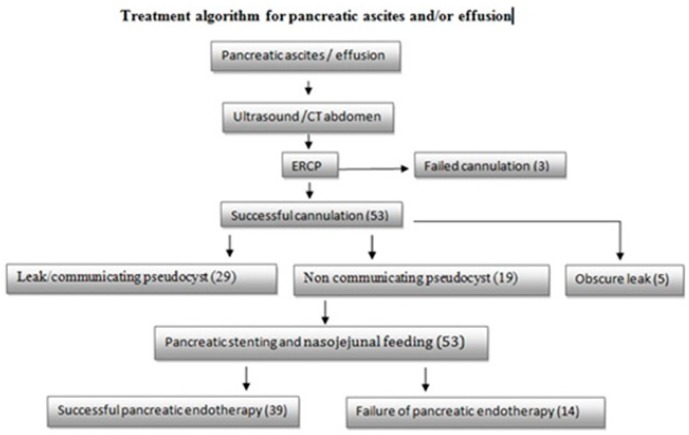
Treatment algorithm. CT: computed tomography; ERCP: endoscopic retrograde cholangiopancreatography.

**Table 1 medsci-05-00006-t001:** Characteristics of the study population.

Total Patients	53
Male:female	7.8:1
Mean age in years	43.89 ± 8.1
Acute pancreatitis	11/53 (20.8%)
Chronic pancreatitis	43/53 (79.2%)
Alcoholic pancreatitis	42/53 (79.2%)
Idiopathic pancreatitis	7/53 (13.2%)
Traumatic pancreatitis	4/53 (7.5%)
Ascites alone	37/53 (69.8%)
Pleural effusion alone	5/53 (9.4%)
Ascites with pleural effusion	11/53 (20.8%)
Mean fluid amylase level IU/mL	5617.21 ± 3311.6

**Table 2 medsci-05-00006-t002:** Univariate analysis of variables affecting the outcome of pancreatic endotherapy at 6 weeks.

Variables		Successful Endotherapy	Fail Endotherapy	*p* Value
Clinical presentation	Ascites	27	10	0.307
	Ascites with effusion	7	4	
	Effusion	5	0	
Type of pancreatitis	Acute pancreatitis	10	1	0.144
	Chronic pancreatitis	29	13	
Etiology of pancreatitis	Alcohol	29	13	0.300
	Traumatic	4	0	
	Idiopathic	6	1	
Leak site	Body	9	1	0.008 *
	Genu	4	0	
	Tail	1	5	
	Pseudocyst	22	6	
	Obscure	3	2	
Stent position	Stent across the leak	27	2	0.0001 *
	Not crossed	8	4	
	Sphincterotomy	1	6	
	Stent placed empirically	3	2	

* = *p* < 0.05 significant.

**Table 3 medsci-05-00006-t003:** Multivariate logistic regression for the variables affecting the outcome of pancreatic endotherapy at 6 weeks.

	Odds Ratio	Standard Error	95% Confidence Interval	*p* Value
Etiology of pancreatitis	0.3269185	0.4043291	0.0289523–3.691441	0.366
Cause of pancreatitis	0.4471668	0.2271196	0.1652485–1.210045	0.113
Type of pancreatitis	1.273519	1.946518	0.0636769–25.47003	0.874
Leak site	2.637887	1.21601	1.068735–6.510916	0.035 *
Stent aross the leak	3.007118	1.163186	1.408953–6.418071	0.004 *

* = *p* < 0.05 significant.

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
