# Peer review of "Efficacy of Pancreatic Endotherapy In Pancreatic Ascites And Pleural Effusion"

_medsci, 2017, doi:10.3390/medsci5020006_

Round 1

Reviewer 1 Report

The authors report a large series focusing on a rare disease (pancreatic ascites)

The study is interesting but data not well explained

Minor suggestions

Please clarify the type of chronic pancreatitis (Calcific? Tropical?) in 43 cases

Please clarify which type of stents were placed (diameter, length)

Was access through the minor papilla required?

Maior comments

Please expand the method section including the ERCP technique

Was MRCP performed in these patients? was it useful to plan the treatment? please add the role of imaging technique (MRCP CTscan) in the discussion section

Data that need to be clarified:

Some examples: 

- From the table seems that the stent was placed in all the 53 patients but in the text it is not clear because it seems that 7 patients received pancreatic sphincterotomy alone

- The leak was identified in 20 patients and 29 received the stent beyond the leak including patients with communicating pseudocyst (the reader do not understand if the leak was seen in 20 or 29 patients).

Pseudocyst communicating with the pancreatic duct do not result in pancreatic ascites and maybe need to be excluded from the series 

Differentiating pancreatic ascites from pseudocysts is important in your series for a better analysis

Author Response

Reply to the Queries On Submitted manuscript entitled “
Efficacy of Pancreatic Endotherapy in Pancreatic Ascites and Pleural Effusion.”

Manuscript ID:

medsci-172595

Thank you for your comments. We have tried to answer the following queries according to our capabilities. Hoping for your needful consideration.

Minor suggestions

Please clarify the type of chronic pancreatitis (Calcific? Tropical?) in 43 cases

Answer=Amongst the 43 patients with chronic pancreatitis, alcohol induced chronic pancreatitis was the commonest cause f/b idiopathic chronic pancreatitis.There were no patients of Tropical pancreatitis in our study, as tropical pancreatitis is common n south India and Nagpur belongs to central india: Where the tropical pancreatitis is rare.

Please clarify which type of stents were placed (diameter, length)

Answer= Plastic pancreatic stents either straight end or single pigtail stents of size 5 fr x 5 cm, 5fr x 7 cm,7frx5 cm were used depending on the site of leak in pancreatic duct.

Was access through the minor papilla required?

Answer=Access through the minor papilla was not required in our patients.

Maior comments

Please expand the method section including the ERCP technique.

Answer (lines 69-81)= In the ERCP setting pancreatic ductal cannulation was achieved with 0.032”guidewire under direct fluoroscopic vision. Pancreatogram was taken to know the presence and site of the leak (Figure 1.) along with associated stones and strictures. After knowing the leak site  plastic pancreatic stent was placed across the leak.Plastic pancreatic stent either straight end or single pigtail stent of size 5 fr x 5 cm, 5fr x 7 cm,7frx5 cm were used depending on the site of leak.Nasojejunal tube feeding was started. Pancreatic stent was removed after 4-6 weeks depending on the time of resolution of ascites/pleural effusion.

Was MRCP performed in these patients? was it useful to plan the treatment? please add the role of  imaging technique (MRCP CTscan) in the discussion section

Answer= CT Abdomen was done in all patients as this was freely available in our setup.MRCP was also done in some of the selected patients due to financial constraints. But not done in all patients especially who had tense ascites due to the possible artifact due to ascetic fluid.Yes we planned the treatment according to the findings on imaging modality. We have focused now on the role of MRCP CT scan in discussion section (lines 153-157):

The advent of helical computed tomography has allowed us to detect contiguous fluid collections and any upstream dilation behind a stricture or stone. Further, imaging with magnetic resonance cholangiopancreatography (MRCP) allows us to take a pancreatogram and give secretin to define the exact site of the disruption.( Richard A. Kozarek Management of Pancreatic Ascites Gastroenterology & Hepatology Volume 3, Issue 5 May 2007)

Data that need to be clarified:

Some examples: 

- From the table seems that the stent was placed in all the 53 patients but in the text it is not clear because it seems that 7 patients received pancreatic sphincterotomy alone.

The leak was identified in 20 patients and 29 received the stent beyond the leak including patients with communicating pseudocyst (the reader do not understand if the leak was seen in 20 or 29 patients).

Answer=Direct leak was demonstrated on pancreatogram in 20 patients.We have also included patients with communicating pseudocyst in the category of leak hence the number was 29.

Pseudocyst communicating with the pancreatic duct do not result in pancreatic ascites and maybe need to be excluded from the series 

Differentiating pancreatic ascites from pseudocysts is important in your series for a better analysis

Answer= We have included patients with pseudocyst demonstration on imaging and who concomitantly also had pancreatic ascites and are analysed accordingly.Sir if u wish us to separate the patients with pseudocyst + ascites from patients with pancreatic ascites only then we can analyse them better  but the number will reduced hence we have combined them together.

In the setting of chronic pancreatitis, leakage is seen in up to 80% of cases from a communicating pseudocyst due to ductal stricture and less commonly (20%) due to duct disruption itself.

(Bracher GA, Manocha AP, DeBanto JR, Gates LK, Slivka A, Whitcomb DC, et al. Endoscopic pancreatic duct stenting to treat pancreatic ascites. Gastrointest Endosc 1999; 49: 710-5.)

Reviewer 2 Report

1.       How is it possible to define a leak site in 20/53 patients but to stent “beyond leak” in 29/53 patients?

2.       Figure 1 is poor quality and uninterpretable to the reviewer. Do you have better quality images?

3.       This is a retrospective review of a database, not a prospective trial. Please redefine.

4.       It is difficult for the reviewer to accept that ¾ of patients “benefited” from endotherapy when timing of endoscopic intervention relative to conservative treatment and underlying ductal changes noted at ERCP are not defined. How many patients had downstream strictures, stones, etc.?

5.       Please comment on complications, if any, related to endotherapy. Please comment on additional therapies that may have influenced “benefit” in addition to endotherapy.

6.       Please discuss reasons for the 3 failures for ERCP.

7.       Here is the major issue: A disconnected pancreatic duct without demonstration of an entire pancreatogram at E/MRCP with high amylase pleural effusion or ascites is likely to come from the upstream duct. A transpapillary stent, then, adds little. Moreover, the endotherapy did not occur in a vacuum – how many patients had TPN/TJ feeding? Octreotide or somatostatin? Diuretics? Large volume paracentesis or thoracentesis? Needs comment in the Discussion section.

Author Response

Reply to the Queries On Submitted manuscript entitled “Efficacy of Pancreatic Endotherapy in Pancreatic Ascites and Pleural Effusion.”

Manuscript ID: medsci-172595

1.    How is it possible to define a leak site in 20/53 patients but to stent “beyond leak” in 29/53 patients?

Answer=Direct leak was demonstrated on pancreatogram in 20 patients.We have also included 9 patients with communicating pseudocyst. Hence the number was 29. As per your suggestion we can separate them and rewrite the result as 20 patients of pancreatic ductal leak were stented beyond the site of leak.We initially categorized communicating pseudocyst in the category of the leak as pancreatogram was showing flow of contrast material into the pseudocyst cavity and CT scan of abdomen has already shown the presence of pseudocyst.

2.    Figure 1 is poor quality and uninterpretable to the reviewer. Do you have better quality images?

Answer= We are submitting the better quality pancreatogram images.

3.    This is a retrospective review of a database, not a prospective trial. Please redefine.

Answer=This was the retrospective database review which was analysed propectively for the preparation of this manuscript.

4.    It is difficult for the reviewer to accept that ¾ of patients “benefited” from endotherapy when timing of endoscopic intervention relative to conservative treatment and underlying ductal changes noted at ERCP are not defined. How many patients had downstream strictures, stones, etc.?

Answer=The definition of “Benefit” in our study was predefined as absence of ascites and/or pleural effusion at week 6 after ductal stenting.This might be a short term definition so as to have benefited 73.6% patients. Almost 70% patients had already received the conservative line of treatment in the form of octreotide,nasojejunal feeding,antibiotics for at leat  4 week ,either in our hospital or other referral centre. However we accept the fact that exact duration of conservative management is variable in our patients. We were able to find out pancreatic ductal stricture in 16 (37.66%) patients among chronic pancreatitis patients and ductal calculi were seen in 21 (48.83%) patients.The facility of Endoscopic ultrasound is not available in our institute hence the exact percent of stones and strictures was not calculated.

5.    Please comment on complications, if any, related to endotherapy. Please comment on additional therapies that may have influenced “benefit” in addition to endotherapy.

Answer = Total 6/53(11.3%) patient died within 6 weeks of enrollment due to progression of underlying ascites and complications like sepsis and bleeding, which occurred in 2 patients.

6.       Please discuss reasons for the 3 failures for ERCP.

Answer= Failure to cannulate even after repeated attempts was the cause as two patients had large juxtapapillary diverticili with papilla at its edge.

7.       Here is the major issue: A disconnected pancreatic duct without demonstration of an entire pancreatogram at E/MRCP with high amylase pleural effusion or ascites is likely to come from the upstream duct. A transpapillary stent, then, adds little. Moreover, the endotherapy did not occur in a vacuum – how many patients had TPN/TJ feeding? Octreotide or somatostatin? Diuretics? Large volume paracentesis or thoracentesis? Needs comment in the Discussion section.

Answer=None of our patient had complete duct disruption syndrome. Most of our patients had one or two leaks with intact remaining pancreatic duct. So this leak was bridged by stent allowing the leak to heal naturally and nasojejunal feed was continued for initial few days till patient tolerated oral feed.

We aim to share our experience from the cohort of patients with pancreatic ductal leak who were stented successfully. Around 70% to 80% patients had already received the conservative line of treatment in the form of octreotide,antibiotics and nasojejunal feeding for at least 4 weeks prior to stenting.No any patient in our study has received Total parenteral Nutrition due to economical constraints. We are working in a  Government hospital in developing country where this was not affordable for poor patients.

 Patients who had massive ascites/pleural effusion causing mechanical distress underwent Therapeutic paracentesis. None of our patient received diuretics considering its less efficacy in pancreatic ascites as compared to High SAAG ascites due to chronic liver disease.

Round 2

Reviewer 2 Report

The manuscript has improved. The reviewer continues to have concerns about the presentation.

The images remain poor. If you do not have better images because of the quality of your C-arm, I suggest adding arrows to the leak site and end of the stent.

The reviewer remains uncertain how you defined the site of the leak when only 20/53 patients had a leak demonstrable on ERCP. How many leak sites were definitively shown with secretin-MRCP? How many were presumptive leak sites based upon the presence of a pseudocyst? Please make this clear in the text or in a Table. This is particularly important as you claim that 29/53 patients had stents placed beyond the leak site.

Figure 2 suggests that 48 patients in this series had a pseudocyst as opposed to 28/53 patients described in the Results section. Please resolve this discordance.

5 and 7 cm stents are not long enough to stent a mid or distal PD body leak. Comment is required in the Discussion section.

You have now included complications and mortality in your revision. Were there no ERCP- related complications? Pancreatitis or its exacerbation? Iatrogenic infection? What was the etiology of GI bleeding that you describe?

Here is the major problem with this retrospective review: Without a control group, it is impossible to define how many of these patients improved because of stenting as opposed to 4–6 weeks of additional conservative management. This needs elaboration in the Discussion section.

Author Response

Reply to the Queries On Submitted manuscript entitled “Efficacy of Pancreatic Endotherapy in Pancreatic Ascites and Pleural Effusion.”

Manuscript ID: medsci-172595

The manuscript has improved.

The reviewer continues to have concerns about the presentation.

1] The images remain poor. If you do not have better images because of the quality of your C-arm, I suggest adding arrows to the leak site and end of the stent.

Answer=Among the available images of pancreatic endotherapy/leak we have submitted the better quality and clear images.We are now making arrows at the point of pancreatic ductal leak and pancreatic stent.

2] The reviewer remains uncertain how you defined the site of the leak when only 20/53 patients had a leak demonstrable on ERCP. How many leak sites were definitively shown with secretin-MRCP? How many were presumptive leak sites based upon the presence of a pseudocyst? Please make this clear in the text or in a Table. This is particularly important as you claim that 29/53 patients had stents placed beyond the leak site.

Answer= Direct leak was demonstrated on pancreatogram in 20 patients.We have also included 9 patients with communicating pseudocyst. Hence the number was 29 of total leak. The site of leak was confirmed by the direct demonstration of  leak on pancreatogram.Prior CT scan was done in all patients however MRCP was done in only few patients because of the financial constraints. We being in the Government medical college ,are not having the facility of free Secretin enhanced MRCP ,hence it was not done.Presumptive i.e. Obscure leak site was present in 5 patients. It has been added in the manuscript.

3] Figure 2 suggests that 48 patients in this series had a pseudocyst as opposed to 28/53 patients described in the Results section. Please resolve this discordance.

Answer=Figure 2 mentions leak and communicating pseudocyst as a group in 29 patients.Out of this 20 patients had direct leak on pancreatogram while 9 patients had communicating pseudocyst.Non communicating pseudocyst was present in 19 patients.Hence total number of patients having pseudocyst is 19+9=28. It has been added in the manuscript.

4] 5 and 7 cm stents are not long enough to stent a mid or distal PD body leak. Comment is required in the Discussion section.

Answer=Our aim was to bridge the pancreatic ductal leak.We have used 125px plastic PD stent for the leak in head region and used 7 fr in body and tail leak.Commonest site of leak in our study was head of pancreas.Yes,For reaching upto the tail end of pancreatic duct more longer stent will be required,but we were able to cross the leak with above stents.

5]You have now included complications and mortality in your revision. Were there no ERCP- related complications? Pancreatitis or its exacerbation? Iatrogenic infection? What was the etiology of GI bleeding that you describe?

Answer= Total 6/53(11.3%) patient died within 6 weeks of enrollment due to progression of underlying ascites and complications like sepsis and bleeding, which occurred in 2 patients.The cause of bleeding was pseudoaeurysm of gastroduodenal  artery ..but patient succumbed before the coiling. There was no ERCP related pancreatitis however underlying disease progressed in 4 patients even after stenting.

6] Here is the major problem with this retrospective review: Without a control group, it is impossible to define how many of these patients improved because of stenting as opposed to 4–6 weeks of additional conservative management. This needs elaboration in the Discussion section.

Answer= The definition of “Benefit” in our study was predefined as absence of ascites and/or pleural effusion at week 6 after ductal stenting.This might be a short term definition so as to have benifitted 73.6% patients.We accept the fact that there is no control group but our aim was to share  our experience regarding the efficacy of pancreatic endotherapy ,how many patients benifitted and what were the factors for the difference in the outcome of endotherapy like site of leak and ability to bridge the leak by stenting. We have added the following paragraph: Absence of the internal control group can be a limitation of our study. However, our prime aim was to share our experience regarding the efficacy of pancreatic endotherapy, how many patients benefited and what were the factors for the difference in the outcome of endotherapy like site of leak and ability to bridge the leak by stenting.

Round 3

Reviewer 2 Report

The authors have addressed the reviewer’s concerns. The images are improved. I presume that there were no ERCP-related complications as the authors have discussed mortality in the series only.

Author Response

Reply to the Queries On Submitted manuscript entitled “Efficacy of Pancreatic Endotherapy in Pancreatic Ascites and Pleural Effusion.”

Manuscript ID: medsci-172595

The authors have addressed the reviewer’s concerns. The images are improved. I presume that there were no ERCP-related complications as the authors have discussed mortality in the series only.

Answer = In our study total 6/53(11.3%) patient died within 6 weeks of enrollment due to progression of underlying ascites and complications like sepsis and bleeding, which occurred in 2 patients.There were no ERCP related complications like post ERCP pancreatitis,this could be because of the smaller number of cases in our series.
